# OpenReview forum: "Learning from Mistakes via Cooperative Study Assistant for Large Language Models"
_EMNLP/2023/Conference — EMNLP 2023 Main_

### Official Review · Reviewer_Qwpe · 2023-08-04

**Typos Grammar Style And Presentation Improvements:** 1. In line 227, “by H).” should be “b…
**Soundness:** 3

**Excitement:**

4: Strong: This paper deepens the understanding of some phenomenon or lowers the barriers to an existing research direction.

**Paper Topic And Main Contributions:**

This paper introduces a new framework SALAM to aid LLMs in their reasoning process. Specifically, compared to previous methods such as self-reflection lacking global insight, the new method can collect a small number of error cases, analyzing the common error and providing solid guidelines for LLMs to solve the new queries. In addition, SALAM’s efficacy and its design choices are empirically proved through comprehensive experiments on subtasks of BBH and BBQ datasets.

**Questions For The Authors:**

    1. Since the model-agnostic study assistant is distilled from GPT4 according to line 379, it would be interesting to use GPT-4 as the study assistant and see the performance upper bound of the SALAM framework.

**Reasons To Accept:**

    1. The related works that the paper cites are up-to-date, thorough, and high-quality from top conferences. The paper is complete and generally easy to follow.
    2. The experiments in this paper are thorough, providing solid support for its claims and research questions.
    3. The proposed generic framework that is based on the model-agnostic study assistant can cast a positive influence on developing efficient methods for mitigating bias and hallucination problems of LLMs.

**Reasons To Reject:**

    1. The description of section 3.4 **Imitation Learning for Study Assistant** starting from line 289 is a bit blurry. How the trajectories are sampled for imitation learning has not been clearly stated. For (s_t, a_t), how are actions before time t selected? And how are “the various possible actions a_t” obtained?
    2. The new method should be compared with previous SOTA ones such as Self-Refine ([https://arxiv.org/abs/2303.17651](https://arxiv.org/abs/2303.17651)) to further show its effectiveness.
    3. The maximum number of actions that SALAM is allowed to take is not specified in the experiments. And there is no ablation study either.

**Reproducibility:**

4: Could mostly reproduce the results, but there may be some variation because of sample variance or minor variations in their interpretation of the protocol or method.

**Reviewer Confidence:**

4: Quite sure. I tried to check the important points carefully. It's unlikely, though conceivable, that I missed something that should affect my ratings.

---

> ### Author Rebuttal · Authors · 2023-08-28
>
> Thank you for your constructive comments! We will polish our manuscript based on your suggestions. Here we would like to first address some concerns.
>
> **Q1. Details about imitation learning**
>
> A1. We detailed the hyper-parameters of imitation learning in Section 4.2 Experiment Setup, especially Line 381-385. Specifically, we sampled 20 trajectories for each mistake of M with a temperature of 0.8.
>
> Besides, for each step t in the sampling, we sample 20 actions based on current state s_t and current policy T. These **various possible actions a_t** are different textual feedback sampled based on Equation 1 in Line 266. For each a_t, we calculate the reward R(s_t, a_t). We stop the sampling of one trajectory if it gets a reward of 1 (correct response). Otherwise, we add the incorrect response to the mistake collection and retrieve a new context s_(t+1). Note that the maximum number of actions depends on the number of options in the query. For example, for one query with 4 options, the maximum number of actions is T=4 because it should arrive at the right answer after 3 failures. We will add all these details in our revision.
>
> ---
>
> **Q2. Comparison with Self-Refine**
>
> A2. We compared our methods with Self-refine (see Table below). We used the same models for generation, feedback, and refinement via different prompts and in-context learning examples. We followed the implementation of their official repo and adapted it to the BBH and BBQ benchmarks. For each benchmark, we use 3 in-context examples for each module. We set the number of iterations to a fixed number (k=3) since the ground truth labels are not accessible during inference.
>
> |                 | BBH  | BBQ  |
> | --------------- | ---- | ---- |
> | M=Flan T5 11B   | 42.4 | 76.6 |
> | Self-refine     | 17.4 | 28   |
> | SALAM w/ replay | 49.0 | 86.4 |
>
> |                 | BBH  | BBQ  |
> | --------------- | ---- | ---- |
> | M=LLaMA 7B      | 26.1 | 24.7 |
> | Self-refine     | 9.8  | 12.4 |
> | SALAM w/ replay | 30.4 | 37.3 |
>
> |                 | BBH  | BBQ  |
> | --------------- | ---- | ---- |
> | M=GPT2 Neox 20B | 24.9 | 26.0 |
> | Self-refine     | 23.8 | 32.5 |
> | SALAM w/ replay | 28.8 | 33.5 |
>
> The above table verifies SALAM achieves better performance than Self-refine. We found that Self-refine’s feedback module can hardly identify the correctness of the current response without knowing the ground truth and it keeps refining the correct answer. It is consistent with our observation as stated in Line 34-39. Besides, it is difficult for less powerful LLMs to reason and create textual feedback only with limited in-context examples.
>
> For example, there is one case from Flan-t5 + Self-refine on BBH which just copied the query.
>
> ```
> Sentence: The designer called the janitor and asked him to clean the room.
> Options:
> (A) Asked the designer
> (B) Asked the janitor
> (C) Ambiguous
> The answer is The designer
> Why is this answer wrong?
>
> Feedback: The designer asked him to clean the room.
> ```
>
> ---
>
> **Q3. The maximum number of actions that SALAM allowed to take**
>
> A3. We would like to clarify the difference between the training and inference. For the training phase, please refer to the response to Q1. For the inference phase, as described in Line 256-259, SALAM will **only take one action**: based on the retrieved context (which includes trajectories with multiple actions) and policy learned from the successful trajectories to provide feedback. It is because the study assistant is not aware of the ground truth during the inference, so it cannot identify whether the response is acceptable or not and continues or not. This is different from training where the study assistant can take action until the LLM gets the correct answer.
>
> ---
>
> **Q4. GPT-4 as the study assistant**
>
> A4. Due to the cost (around 1000$ for each LLM), we used GPT-4 (gpt-4-0613) to provide feedback on a small subset (10%) of the training set for M = LLaMA and presented the results below. For a fair comparison, we also provide SALAM with 10% feedback as one baseline. From the table, we can see that given the 10% feedback, GPT-4 outperformed SALAM by 2.1 on BBH and 0.7 on BBQ. However, SALAM with 100% feedback outperformed GPT-4, highlighting the importance of diverse feedback. Since our SALAM is much cheaper than GPT-4, it demonstrates the potential of our SALAM to provide feedback.
>
> | M=LLaMA                 | BBH  | BBQ  |
> | ----------------------- | ---- | ---- |
> | T = 10% GPT-4           | 26.9 | 30.9 |
> | SALAM w/ 10 % feedback  | 24.8 | 30.2 |
> | SALAM w/ 100 % feedback | 28.7 | 34.9 |

---

### Official Review · Reviewer_5R6j · 2023-08-04

**Soundness:** 3

**Excitement:**

4: Strong: This paper deepens the understanding of some phenomenon or lowers the barriers to an existing research direction.

**Paper Topic And Main Contributions:**

This paper focuses on the ability of language models to refine their output based on some feedback and faces the challenge of the correctness of the feedback. The designed framework improves the reflection and refinement process of large language models, by grading previous responses given the ground truth and collecting mistakes. The suggested solution is evaluated on two benchmarks, Big-Bench-Hard and Bias Benchmark for QA and compared to other baselines.

**Questions For The Authors:**

- The paper could be improved by comparing their method against one or more similar methods on the same dataset/benchmark. One such method could be self-refinement.
- It would be nice to see the learning curves of the trained model that provides feedback and provide more details about the algorithm and the parameters.
- It would be interesting to see how Llama 2 performs as it is pre-trained on much more data than Llama.

**Reasons To Accept:**

- This work proposes a novel method for reflection and refinement which can be impactful on many application.
- The experimental results showcase the effectiveness of the proposed method.

**Reasons To Reject:**

- The experimental setup is not clearly described in the document, which might be an issue if someone wants to replicate the experiments. The authors claim that code will be provided, however more details about the experiments and the configurations should be provided in Appendix.
- The authors compare their method against some baselines but it is not clear how their method stands against other approaches such as Madaan et al. 2023 or others.

**Reproducibility:**

5: Could easily reproduce the results.

**Reviewer Confidence:**

3: Pretty sure, but there's a chance I missed something. Although I have a good feel for this area in general, I did not carefully check the paper's details, e.g., the math, experimental design, or novelty.

---

> ### Author Rebuttal · Authors · 2023-08-28
>
> Thank you for your detailed and valuable comments! Hope our detailed replies can solve your concerns.
>
> **Q1 The code and detailed experimental configuration.**
>
> A1. We kindly refer the reviewer to the supplementary materials where we uploaded our code for reproduction. Besides, we described the experimental configuration in Section 4.2 and hyper-parameters in Line 385-389. Specifically, we finetuned all models on two A6000 GPUs for 10 epochs with a learning rate of 2e-5 and updated every 32 instances. It took about 7 hours to fine-tune one model. These hyper-parameters are used for all models (including the SALAM and SALAM w/ replay). The other settings are the same as the original LLaMA. For other details about the fine-tuning data, we use the prompt in Figure 2 to distillate GPT-4 for the feedback data (514 for BBQ and 1855 for BBH). We will release this feedback dataset. We will revise our manuscript to make it more clear.
>
> ---
>
> **Q2. Self-refine baseline (Madaan et al. 2023)**
>
> A2. We compared our methods with Self-refine (see Table below). We used the same models for generation, feedback and refinement via different prompts and in-context learning examples. We followed the implementation of their official repo and adapted it to the BBH and BBQ benchmarks. For each benchmark, we use 3 in-context examples for each module. We set the number of iterations to a fixed number (k=3), since the ground truth labels are not accessible during inference.
>
> |                 | BBH  | BBQ  |
> | --------------- | ---- | ---- |
> | M=Flan T5 11B   | 42.4 | 76.6 |
> | Self-refine     | 17.4 | 28   |
> | SALAM w/ replay | 49.0 | 86.4 |
>
> |                 | BBH  | BBQ  |
> | --------------- | ---- | ---- |
> | M=LLaMA 7B      | 26.1 | 24.7 |
> | Self-refine     | 9.8  | 12.4 |
> | SALAM w/ replay | 30.4 | 37.3 |
>
> |                 | BBH  | BBQ  |
> | --------------- | ---- | ---- |
> | M=GPT2 Neox 20B | 24.9 | 26.0 |
> | Self-refine     | 23.8 | 32.5 |
> | SALAM w/ replay | 28.8 | 33.5 |
>
> The above table verifies SALAM achieves better performance than Self-refine. We found that Self-refine’s feedback module can hardly identify the correctness of the current response without knowing the ground truth and it keeps refining the correct answer. It is consistent with our observation as stated in Line 34-39. Besides, it is difficult for less powerful LLMs to reason and create textual feedback only with limited in-context examples.
>
> For example, there is one case from Flan-t5 + Self-refine on BBH which just copied the query.
>
> ```
> Sentence: The designer called the janitor and asked him to clean the room.
> Options:
> (A) Asked the designer
> (B) Asked the janitor
> (C) Ambiguous
> The answer is The designer
> Why is this answer wrong?
>
> Feedback: The designer asked him to clean the room.
> ```
>
> ---
>
> **Q3. Learning curves of the feedback model**
>
> A3. Thanks for your advice and we will add this information in our appendix. Due to the rebuttal policy, we cannot provide an external link for the figure of the learning curve. Here we briefly show the training loss of different steps in the table below. The loss converged after 150 steps on both datasets.
>
> | Step | BBH   | BBQ   |
> | ---- | ----- | ----- |
> | 25   | 7.438 | 7.407 |
> | 50   | 2.314 | 2.710 |
> | 75   | 0.045 | 0.067 |
> | 100  | 0.014 | 0.006 |
> | 125  | 0.009 | 0.004 |
> | 150  | 0.006 | 0.002 |
> | 175  | 0.007 | 0.003 |
> | 200  | 0.007 | 0.002 |
>
> ---
>
> **Q4. The performance of LLaMA-2**
>
> A4. We chose LLaMA-1 instead of LLaMA-2 because LLaMA-2 hadn't been released at that time. Due to the time limitation, we will add the experiments in our revision.

---

### Official Review · Reviewer_hBhU · 2023-08-16

**Soundness:** 4

**Excitement:**

4: Strong: This paper deepens the understanding of some phenomenon or lowers the barriers to an existing research direction.

**Missing References:**

Some of the relevant references in "Language Models as Feedback" *might* be missing. For instance MemPrompt [1] proposed a very related idea, to fetch relevant feedback using a database of <error, user feedback> pairs, but was not discussed or cited.

**References**
1. Madaan, A., Tandon, N., Clark, P., & Yang, Y. (2022). Memory-assisted prompt editing to improve GPT-3 after deployment. ArXiv, abs/2201.06009.

**Paper Topic And Main Contributions:**

This paper addresses the problem of helping large language models (LLMs) learn from their mistakes to improve their reasoning and bias reduction abilities. The main contribution is a framework called SALAM that introduces a "study assistant" agent implemented as an LLM. The study assistant analyzes mistakes made by the LLM on a training set, identifies common misunderstandings, and provides feedback to help avoid similar mistakes at the inference. It does this by retrieving similar mistakes from a collected set during training. SALAM is model-agnostic but can also be tailored to a specific LLM's behavior using imitation learning on the training set. The authors perform extensive experiments on 27 tasks across 3 different models, demonstrating decent performance improvements.

**Questions For The Authors:**

D. **Computational Overhead of Imitation Learning for Study Assistant:** What is the computational overhead of using imitation learning to tailor the study assistant to individual models compared to more standard supervised finetuning or other optimization strategies? Some analysis of relative costs would be informative.

E. **Comparison to Supervised Finetuning Baseline:** Given that SALAM requires access to training data, it would be useful to compare performance to a supervised finetuning baseline trained on the full dataset. Although not a fair comparison, the SFT method would rather act as a gold standard for the proposed SALAM. This could offer practical insights into when SALAM is preferred over finetuning given sufficient data and compute.

F. **SALAM with 100% Training Set:** Expanding on the data efficiency results, how does SALAM perform with 100% of training data versus the 10% subset? This could reveal how much data is needed for SALAM to be maximally effective.

**Reasons To Accept:**

The proposed SALAM method is a good step forward in improving reasoning capabilities of LLMs, through the use of another LLM that provides relevant feedback. The authors have shown decent performance improvements on various tasks, and also perform various analysis and ablations to dissect the importance of various components. Further, the proposed method can be used in a model-agnostic fashion with probably large LLMs. In summary, the proposed method is a good contribution to the community for improving the reasoning capabilities of LLMs.

**Reasons To Reject:**

I do not find any major weaknesses that would preclude acceptance of this paper. However, several additions could further strengthen the work:

A. **Baselines:** The current method is not compared with any baselines from previous methods in literature. Introducing baselines like Self-Refine [1] or Self-Correct [2] would provide useful context on how SALAM compares.

B. **Study-Assistant Feedback for Large Models:** It would be insightful to evaluate SALAM's ability to provide effective feedback for much larger LLMs, like GPT-3 or GPT-4, using the smaller study assistant model. This could reveal benefits or limitations compared to pairing study assistant and student models of similar scale.

C. **Detailing of Points in Appendix:** Some details are excluded from the main text and supplementary material, such as how the LLaMA study assistant was finetuned using GPT-4 data. Providing SALAM's performance on individual tasks (with 100% training data) would also be helpful.

**References**
1. Madaan, A., Tandon, N., Gupta, P., Hallinan, S., Gao, L., Wiegreffe, S., Alon, U., Dziri, N., Prabhumoye, S., Yang, Y., Welleck, S., Majumder, B.P., Gupta, S., Yazdanbakhsh, A., & Clark, P. (2023). Self-Refine: Iterative Refinement with Self-Feedback. ArXiv, abs/2303.17651.
2. Welleck, S., Lu, X., West, P., Brahman, F., Shen, T., Khashabi, D., & Choi, Y. (2022). Generating Sequences by Learning to Self-Correct. ArXiv, abs/2211.00053.

**Reproducibility:**

5: Could easily reproduce the results.

**Reviewer Confidence:**

4: Quite sure. I tried to check the important points carefully. It's unlikely, though conceivable, that I missed something that should affect my ratings.

---

> ### Author Rebuttal · Authors · 2023-08-28
>
> We appreciate your positive feedback and detailed comments! We will add the missing reference and more experimental details in our manuscript. We also prepare detailed responses to address other concerns.
>
> **Q1. baselines such as Self-refine or Self-correct**
>
> A1. We compared our methods with Self-refine (see Table below). We used the same models for generation, feedback, and refinement via different prompts and in-context learning examples. We followed the implementation of their official repo and adapted it to the BBH and BBQ benchmarks. For each benchmark, we use 3 in-context examples for each module. We set the number of iterations to a fixed number (k=3) since the ground truth labels are not accessible during inference.
>
> |                 | BBH  | BBQ  |
> | --------------- | ---- | ---- |
> | M=Flan T5 11B   | 42.4 | 76.6 |
> | Self-refine     | 17.4 | 28   |
> | SALAM w/ replay | 49.0 | 86.4 |
>
> |                 | BBH  | BBQ  |
> | --------------- | ---- | ---- |
> | M=LLaMA 7B      | 26.1 | 24.7 |
> | Self-refine     | 9.8  | 12.4 |
> | SALAM w/ replay | 30.4 | 37.3 |
>
> |                 | BBH  | BBQ  |
> | --------------- | ---- | ---- |
> | M=GPT2 Neox 20B | 24.9 | 26.0 |
> | Self-refine     | 23.8 | 32.5 |
> | SALAM w/ replay | 28.8 | 33.5 |
>
> The above table verifies SALAM achieves better performance than Self-refine. We found that Self-refine’s feedback module can hardly identify the correctness of the current response without knowing the ground truth and it keeps refining the correct answer. It is consistent with our observation as stated in Line 34-39. Besides, it is difficult for less powerful LLMs to reason and create textual feedback only with limited in-context examples.
>
> For example, there is one case from Flan-t5 + Self-refine on BBH which just copied the query.
>
> ```
> Sentence: The designer called the janitor and asked him to clean the room.
> Options:
> (A) Asked the designer
> (B) Asked the janitor
> (C) Ambiguous
> The answer is The designer
> Why is this answer wrong?
>
> Feedback: The designer asked him to clean the room.
> ```
>
> For Self-correct, the main idea is to train a separate corrector model. It refines the model output given the query, the previous response and the feedback. Their feedback during the inference comes from gold verifiers, which is not available in our settings.
>
> ---
>
> **Q2. larger LLM such as GPT-3 or GPT-4**
>
> A2. We applied our SALAM to the latest GPT-4 (gpt-4-0613). Specifically, we used GPT-4 as LLM and our SALAM to provide feedback. Due to the high cost ( around 1500$ on two benchmarks), we only conducted the comparison on a random subset (10%) of the original test set. The results revealed that while GPT-4 already exhibited strong performance on the BBQ benchmark, leaving limited room for SALAM to enhance, on BBH SALAM significantly boosted GPT-4's performance. This suggests that even a large model like GPT-4 can benefit from feedback provided by our study assistant.
>
> |         | BBH  | BBQ  |
> | ------- | ---- | ---- |
> | M=GPT-4 | 72.9 | 98.2 |
> | SALAM   | 75   | 98.2 |
>
> ---
>
> **Q3. The detailed experimental settings**
>
> A3. We kindly refer the reviewer to the supplementary materials where we uploaded our code for reproduction. Besides, we described the experimental configuration in Section 4.2 and hyper-parameters in Line 385-389. Specifically, We finetuned all models on two A6000 GPUs for 10 epochs with a learning rate of 2e-5 and a batch size of 32. It took about 7 hours to converge. These hyper-parameters are used for all models (including the SALAM and SALAM w/ replay). The other settings are the same as the original LLaMA.
>
> For other details about the fine-tuning data, we use the prompt in Figure 2 to distillate GPT-4 for the feedback data (514 for BBQ and 1855 for BBH). This feedback dataset will also be released with our code. We will make it more clear in our revision.
>
> ---
>
> **Q4. Computational overhead of imitation learning**
>
> A4. One of the advantages of SALAM is that it can use a small study assistant to guide a large LLM, without the cost of finetuning the LLM. If the LLM has the same size as the study assistant, the computation costs for finetuning and imitation learning are similar (both converge after 150 steps). However, when the LLM is larger than the study assistant, it requires more computational resources.
>
> ---
>
> **Q5. Supervised finetuning baseline**
>
> A5. Here we provide the supervised baseline for LLaMA (7b) finetuned on the same training set. Flan-t5 (11b) and GPT-NeoX (20b) caused OOM even with batch size=1 on the A6000 GPU, which makes it impossible for us to fully fine-tune these models. It also demonstrates the advantage of SALAM which is more computationally efficient. We used the same hyperparameters as the study assistant, and the model converged after 150 steps on both benchmarks. As the results show, the supervised models outperformed other models by a large margin on BBQ, indicating that the social bias can be effectively reduced with the finetuning data. However, for the reasoning benchmark BBH, the supervised model does not have more advantages. We suppose it is because complex reasoning is more difficult to learn with limited data. However, with the assistance of our SALAM, it is easier to figure out the common misunderstanding and can better generalize.
>
> | LLAMA           | BBH  | BBQ  |
> | --------------- | ---- | ---- |
> | LLaMA           | 26.1 | 24.7 |
> | LLaMA w/ SFT    | 29.3 | 74   |
> | SALAM w/ replay | 30.4 | 37.3 |
>
> ---
>
> **Q6. SALAM with 100% Training Set on BBH**
>
> A6. We list the full results on BBH in the following table and will add it to our appendix. The hyper-parameters are the same for 10% and 100%, such as k=3 and theta=0.9.
>
> In Line 472-477 of the main body, we find that SALAM struggles with complex tasks such as tracking shuffled objects and geometric shapes. Here we can see with more data, the performance of SALAM on these tasks improved significantly. However, the performance on some simple tasks degrades. We checked the results and found that it retrieved less relevant examples. Under the same retrieval setting, the larger training set may add some noise to the retrieved context, leading to the same observation as the retrieval analysis in Section 4.5.
>
> |                                 | 10% Data | 100% Data |
> | ------------------------------- | -------- | --------- |
> | date understanding              | 46.0     | 50.0      |
> | disambiguation qa               | 80.0     | 78.0      |
> | geometric shapes                | 14.0     | 28.0    |
> | hyperbaton                      | 84.0     | 88.0      |
> | logical deduction three         | 72.0     | 62.0      |
> | logical deduction five          | 70.0     | 56.0      |
> | logical deduction seven         | 62.0     | 56.0      |
> | movie recommendation            | 42.0     | 72.0      |
> | penguins in a table             | 43.3     | 36.7      |
> | reasoning about colored objects | 64.0     | 64.0      |
> | ruin names                      | 26.0     | 40.0      |
> | snarks                          | 75.0     | 58.3      |
> | temporal sequences              | 26.0     | 24.0      |
> | tracking shuffled objects three | 24.0     | 28.0      |
> | tracking shuffled objects five  | 10.0     | 14.0      |
> | tracking shuffled objects seven | 16.0     | 22.0      |
> | **Average**                     | 47.1     | 48.6      |

---

### Official Review · Reviewer_2bb5 · 2023-08-18

**Soundness:** 3

**Excitement:**

4: Strong: This paper deepens the understanding of some phenomenon or lowers the barriers to an existing research direction.

**Paper Topic And Main Contributions:**

This paper introduces a novel framework, SALAM, to help large language models (LLMs) improve their generation quality by providing feedback based on their previous mistakes. SALAM acts as a study assistant for LLMs, grading their responses with the ground truth and collecting common errors in the training phase. In the inference phase, SALAM guides LLMs to avoid similar mistakes by generating hints based on the error collection.

**Reasons To Accept:**

1: The paper explores how human study assistants help students by detecting common mistakes and offering guidance. This approach is both intriguing and novel. The paper found that failure is more valuable than success, this is pretty interesting. Previously, we just use correct few-shot for LLM inference, but this paper delivered a quite different and interesting idea.

2: The paper defined the problem clearly, described it mathematically and formulated an optimization function.

3: The effectiveness of the proposed method was evidenced by a robust and comprehensive analysis and ablation studies.

**Reasons To Reject:**

1: The SALAM framework relies on the availability and quality of ground truth labels for grading and providing feedback. However, in some domains or tasks, such labels might be scarce, noisy, or subjective. For example, in open-ended text generation or summarization, there might be multiple valid or acceptable outputs that are not covered by the ground truth. How does SALAM handle such cases and ensure that its feedback is not biased or misleading?

2: The SALAM framework assumes that the LLM can understand and follow the feedback provided by the study assistant. However, this might not be always true, especially for complex or abstract feedback that requires reasoning or inference. For example, if the feedback is “avoid using vague or ambiguous terms”, how does the LLM know which terms are vague or ambiguous and how to replace them with more specific ones? How does SALAM evaluate the effectiveness of its feedback and adjust it accordingly?


**Reproducibility:**

3: Could reproduce the results with some difficulty. The settings of parameters are underspecified or subjectively determined; the training/evaluation data are not widely available.

**Reviewer Confidence:**

2: Willing to defend my evaluation, but it is fairly likely that I missed some details, didn't understand some central points, or can't be sure about the novelty of the work.

---

> ### Author Rebuttal · Authors · 2023-08-28
>
> Thank you for your insightful comments! We have carefully considered your comments and also provide detailed responses to address your concerns.
>
> **Q1: How can SALAM deal with missing or unreliable labels?**
>
> A1: Here, we would like to highlight that our framework is generic and can be easily adapted to these scenarios. We will discuss these scenarios respectively.
>
> **For tasks without ground-truth labels or tasks with noisy data**, we can manually annotate a few high-quality examples for them as the ground truth and apply SALAM to learn feedback from them. As we discussed in Section 4.4 and Table 3, SALAM is sample efficient and can outperform baselines with only 10% data (25 examples per task) on BBH. It verifies that the annotation process of SALAM is much cheaper than the traditional finetuning.
>
> **For those open-ended generations,** In our settings, there are multiple options for one query, and the study assistant is used to explain why option A (model output) is incorrect and option B (ground truth) is correct. For open-ended generation,as long as the data can be formulated as multiple choices (one is the preferred and the others are rejected), our SALAM can be adapted to it. The LLM creates initial outputs during the first pass on the training step, and these outputs are used to create feedback and train the study assistant.
>
> **For the potential bias or noise of the ground truth**, we can filter them by manually checking the feedback or creating heuristic rules.
>
>
>
> ---
>
>
>
> **Q2: How can we check the LLM’s understanding of the feedback? How to evaluate the effectiveness of the feedback?**
>
> A2: To evaluate the understanding and effectiveness of the feedback, we investigate the performance of the LLMs on the downstream tasks.
>
> The goal of the feedback is to improve the performance of LLMs on specific tasks. **If the feedback can actually help different LLMs get a better score (in this paper is accuracy), it indicates the feedback is effective.** In Table 2, SALAM helps three basic LLMs achieve the best average performance on BBH and BBQ. **On the other hand, if the LLM can benefit more from the feedback, it indicates that it can better understand the feedback and follow the instructions.** Therefore, the magnitude of the performance gain illustrates the understanding of LLMs. Moreover, the imitation learning (Section 3.4) further enhances the feedback towards one specific LLM, aiming to improve this LLM’s understanding of feedback.

---

### Meta-Review · Area_Chair_UG6E · 2023-09-13

**Recommendation:** 4

**Metareview:**

**Strengths**:

1. SALAM helps improve LLM performance by providing a systematic framework to help the LLM learn from previous mistakes. It collects mistakes in the training phase. During inference, it uses this mistake collections to provide guidelines for LLMs to help them avoid similar mistakes. The idea is novel and intriguing.

2. Authors present a principled framework with all mathematical details.

3. comprehensive analysis (on BBH and BBQ datasets with 3 models) and ablation studies.

4. Code is shared. Feedback dataset will also be released.

**Weaknesses**:


1. For multi-step reasoning, the feedback could be indirect and not targeted. I think more work needs to be done to structure out which part of the reasoning is making mistakes and should be influenced by feedback.

2. "Computational overhead of imitation learning" -- more objectivity (trying out different study assistant sizes and then comparing their relative impact) would be nice to have.

**Suggestions**:

1. Include some learnings from these in main paper: (1) Comparison with baselines like Self-Refine or Self-Correct. (2) expts with larger LLMs like GPT4. (3) Supervised finetuning baseline Llama results. (4) SALAM with 100% Training Set on BBH. (5) GPT-4 as the study assistant

2. Please update the writeup for 3.4 Imitation Learning for Study Assistant .

---

### Decision · Program_Chairs · 2023-10-07

**Decision:**

Accept-Main

**Comment:**

**Strengths**:

1. SALAM helps improve LLM performance by providing a systematic framework to help the LLM learn from previous mistakes. It collects mistakes in the training phase. During inference, it uses this mistake collections to provide guidelines for LLMs to help them avoid similar mistakes. The idea is novel and intriguing.

2. Authors present a principled framework with all mathematical details.

3. comprehensive analysis (on BBH and BBQ datasets with 3 models) and ablation studies.

4. Code is shared. Feedback dataset will also be released.

**Weaknesses**:


1. For multi-step reasoning, the feedback could be indirect and not targeted. I think more work needs to be done to structure out which part of the reasoning is making mistakes and should be influenced by feedback.

2. "Computational overhead of imitation learning" -- more objectivity (trying out different study assistant sizes and then comparing their relative impact) would be nice to have.

**Suggestions**:

1. Include some learnings from these in main paper: (1) Comparison with baselines like Self-Refine or Self-Correct. (2) expts with larger LLMs like GPT4. (3) Supervised finetuning baseline Llama results. (4) SALAM with 100% Training Set on BBH. (5) GPT-4 as the study assistant

2. Please update the writeup for 3.4 Imitation Learning for Study Assistant .